

# Low cost macroalgal canopy biomass monitoring using light attenuation

Mark P. Johnson

School of Natural Sciences and Ryan Institute, University of Galway, Galway, Ireland

## ABSTRACT

Macroalgal canopies are productive and diverse habitats that export material to other marine ecosystems. Macroalgal canopy cover and composition are considered an Essential Ocean Variable by the research community. Although several techniques exist to both directly and remotely measure algal canopies, frequent measures of biomass are challenging. Presented here is a technique of using the relative attenuation of light inside and outside canopies to derive a proxy for algal biomass. If canopy attenuation coefficients are known, the proxy can be converted to an area of algal thallus per seabed area (thallus area index). An advantage of the approach is that light loggers are widely available and relatively inexpensive. Deployment for a year in the intertidal demonstrated that the method has the sensitivity to resolve summertime peaks in macroalgal biomass, despite the inherent variation in light measurements. Relative attenuation measurements can complement existing monitoring, providing point proxies for biomass and adding seasonal information to surveys that sample shores at less frequent intervals.

## INTRODUCTION

Macroalgal canopies support adjacent coastal ecosystems by exporting detritus (*Bustamante & Branch, 1996*; *Vetter & Dayton, 1998*; *Renaud et al., 2015*; *Both et al., 2020*) and supply pools of carbon to sediments (*Queirós et al., 2019*). These ecosystem functions are linked to the productivity and biomass of algal canopies (*Binzer & Sand-Jensen, 2002*), with loss of canopy biomass disrupting coastal food webs (*Byrnes et al., 2011*; *Duarte et al., 2015*). Canopy biomass is affected by a range of factors including grazing activity, harvests, eutrophication and climate change (*Davies, Johnson & Maggs, 2007*; *Strain et al., 2014*; *Carnell & Keough, 2019*; *Filbee-Dexter et al., 2020*; *Gizzi et al., 2021*; *Bularz et al., 2022*). The sensitivity of macroalgal canopies to a number of stressors is reflected in the selection of macroalgal canopy cover and composition as an essential ocean variable (EOV) in the Global Ocean Observing System (GOOS, https://www.goosocean.org). Essential ocean variables are meant to bring clarity and coherence to monitoring programmes, allowing assessments of the state of marine environments and ecosystems. Monitoring macroalgal canopy biomass also feeds into regional and national programmes, such as evaluations of environmental status under the EU Marine Strategy Framework (*OSPAR Commission, 2012*).

Corresponding author
Mark P. Johnson,
mark.johnson@nuigalway.ie

Various methods exist for monitoring macroalgal canopies, not all of which provide an estimate of biomass. *Duffy et al. (2019)* reviewed the patchy global coverage of macroalgal observation programmes, with 15 of the 59 reviewed programmes reported as measuring biomass. Biomass is not necessarily equivalent to widely reported measures like percentage cover. Where a quadrat is placed on a canopy, the maximum value of cover is 100%. A particular challenge for intertidal canopies is that it is difficult for visual methods to discriminate different amounts of biomass that might all be recorded as 100% cover. One or several layers of algae could both be recorded as 100% cover despite wide differences in biomass. *Burrows, Harvey & Robb (2008)* showed that abundance categories based on visual estimates of percent cover have some predictive value in estimating macroalgal biomass (4% $<r^2$ <34%), but these relationships were not tested above 60% cover: where potential layering of fronds becomes more important. Similarly, *Guichard, Bourget & Agnard (2000)* were able to estimate algal biomass using camera-based measurements of normalized difference vegetation index (NDVI), but the calibration range was less than 10% of the maximum algal biomass on the shore. When the canopy is floating, remotely sensed biomass estimates may be easier (*Cavanaugh et al., 2010*; *Hu, Hu & Ming-Xia, 2017*), with satellite data also showing promise for monitoring thinner, more translucent algae like *Ulva* spp. in estuaries (*Karki et al., 2021*). However, visual estimates of biomass from multi-layered canopies of leathery algae like fucoids and kelps on rocky substrates remain challenging.

The most direct methods to estimate macroalgal biomass involve some sort of harvest, most commonly by removing and weighing all fronds within a sample quadrat. This is a destructive process that may impact the monitored habitat. Canopy removal experiments have demonstrated a variety of recovery times, including permanent conversion to a different dominant cover, influenced by factors like removal extent, recruitment variability and location-specific processes (*Jenkins, Norton & Hawkins, 2004*; *Methratta & Petraitis, 2008*; *Menge et al., 2017*). *Johnson (2020)* estimated that between 5 and 36 0.25 m$^2$ quadrats (depending on fucoid species) would be required to generate a site biomass estimate with upper confidence intervals less than twice the mean. The potential impact of harvests for biomass estimation suggests that monitoring using this approach should be carefully evaluated. A technique to reduce the requirement for removal of canopy algae when estimating biomass is to sacrifice a small number of individual fronds and then estimate biomass by multiplication of frond size and estimated density (*e.g.*, *Attard et al., 2019*; *Smale et al., 2020*). This precision of this technique remains to be established as the biomass estimates contain multiplied sampling error from density and biomass estimates, along with any decisions about what the sampled individual fronds represent (*e.g.*, estimates based on canopy forming individuals only).

Monitoring programmes can be difficult to sustain (*Satterthwaite et al., 2021*), resulting in trade-offs over features like spatial and temporal coverage. Over 80% of the macroalgal monitoring programmes collated by *Duffy et al. (2019)* report annual or greater sampling intervals. Monitoring of seasonal variation in canopy biomass is therefore rare. This lack of seasonal information could be an issue as annual monitoring and climate means may not reflect the mechanisms linking environmental variables to ecological responses, reducing
the understanding and prediction of ecosystems affected by climate change (*Helmuth et al., 2014*; *Bates et al., 2018*).

Although quadrat and transect surveys remain important for ground truthing (and dominate the macroalgal monitoring programmes in *Duffy et al. (2019)*, methods for monitoring macroalgae are continually developing, including technologies like satellite and drone sensing (*Schroeder et al., 2019*; *Rossiter et al., 2020*; *Tait, Orchard & Schiel, 2021*), acoustics (*Schimel, Brown & Ierodiaconou, 2020*), LIDAR (*Webster et al., 2020*) and video analysis (*Katz et al., 2021*). The strengths, weaknesses and challenges of various approaches reflect various features of the available technology, see comments by *Miloslavich, Johnson & Benedetti-Cecchi (2018)* and *Bell et al. (2020)*. In particular, the remote sensing methods tend to have limitations in expense and/or expertise required, alongside some specific issues like working around tidal height variation and use in water of high turbidity for some approaches (*Miloslavich, Johnson & Benedetti-Cecchi, 2018*).

There is a gap for a canopy biomass monitoring method which is simple to apply, minimizes destructive harvests and can account for variations in water turbidity. By considering the light that passes through the canopy, it is possible to go beyond the limits of two-dimensional visual estimates of percentage cover. Furthermore, the use of dataloggers allows an increased temporal resolution, demonstrating seasonal changes in biomass and potentially providing insights about causes of variation in canopy biomass. The method presented here uses low-cost light loggers inside and outside canopy algae to estimate the attenuation associated with macroalgae. The extent of attenuation provides a proxy for biomass that can be calibrated using a harvest or, ideally, converted to biomass once reference canopy attenuation coefficients are defined. A potential drawback of using relative attenuation is that underwater light is highly variable, and this 'noise' may obscure any signal associated with changes in macroalgal biomass. This manuscript evaluates whether datalogging can be a suitable source for estimates of macroalgal biomass.

## MATERIALS & METHODS

### Background to use of relative attenuation

The derivation of a proxy for algal biomass is based on the light intercepted by the canopy before the seabed is reached. A Beer-Lambert law approximation for the light on the seabed below a canopy is:

$$I_c = I_s . e^{(-k_w z - k_a T_{AI})} \qquad (1)$$

where $I_c$ is the light at the seabed, $I_s$ is the surface light, $k_w$ is the attenuation coefficient of sea water ($m^{-1}$), $z$ is the depth of water above the seabed (m), and $k_a$ ($T_{AI}^{-1}$) is the attenuation from the area of algal canopy above the seabed ($T_{AI}$ is the thallus area index: $m^2$ algal frond $m^{-2}$ seabed). The units for light depend on the sensor used. Low-cost sensors often measure intensity as lux.

Although $T_{AI}$ is based on a ratio of areas, it is considered in the current study to be a measure of biomass per unit area (as total thallus area is considered to be linearly related to biomass, (*Johnson et al., 1998*; *Mauffrey, Cappelatti & Griffin, 2020*)). This allows $T_{AI}$

to be distinguished from terms like 'cover' or 'areal coverage', which, without further qualification, are ambiguous about how the biomasses of multilayered canopies are described.

For seabed not overshadowed by algal canopy, the light falling on the seabed ($I_o$) can be approximated by:

$$I_o = I_s . e^{(-k_w z)}. \tag{2}$$

The effect of the canopy on light interception can be estimated by comparing $I_c$ and $I_o$. A larger difference between the two implies more canopy attenuation, reflecting higher canopy biomass. If two light sensors are at the same depth and in the same body of water, the $k_w z$ terms will be equivalent in the two equations. Taking natural logarithms of Eqs. (1) and (2) and subtracting (2) from (1) removes the $I_s$, $z$ and $k_w$ terms, resulting in:

$$ln\left(\frac{I_0}{I_c}\right) = k_a T_{AI}. \tag{3}$$

The derivation of Eq. (3) therefore provides an index of algal biomass based on the ratio of canopy and non-canopy sensors. If $k_a$ is known, then the ratio can be interpreted in terms of the thallus area. If $k_a$ is not known, then the quantity $ln (I_o/ I_c)$ can be considered as an index of biomass ('T$_{AI}$ index').

The underwater light climate is known for variability. The attenuation coefficient of water can change (for example with plankton concentration), along with transient peaks of light associated with wave lensing (*e.g.*, *Schubert, Sagert & Forster , 2001*). Under canopies, the degree of light interception will vary with frond overlap, including sun flecks as algae are moved by waves (*Wing & Patterson, 1993*). Light at the seafloor will vary with the depth of the water column. Furthermore, attenuation will be affected by the balance between direct and diffuse light, and changes in attenuation associated with different wavelengths.

Some of the effects of light climate variability on $I_o/ I_c$ will be minimized by placing sensors at the same depth in relative proximity. Under these constraints, sensors will record under the same water depth regardless of tidal or other depth fluctuations and local variations in attenuation coefficient are likely to be minor. Other sources of variability are likely to be amplified, particularly those associated with a moving canopy. The benefit of using data logging sensors in the field is that a high recording rate may allow the signal of algal biomass to be identified amongst the noise. This manuscript presents details of the signal from one year of light measurements recorded by sensors in the intertidal.

## Field observations

Irradiance was logged from 8 HOBO MX2202 temperature/light sensors deployed in the low to mid intertidal at Furbo in Galway (53.246°N, 9.221°W). The tidal range in Galway Bay is approximately 1.2 m on neaps and 5 m on springs and the site at Furbo is moderately exposed to waves. The HOBO loggers were fixed horizontally to the shore using marine putty. The average depth of loggers was estimated to be 1.8 m above chart datum, resulting in them being underwater approximately 80% of the time, with an average of 1.4 m of water depth, maximum 4 m. This location contains a patchy canopy of *Fucus vesiculosus* and

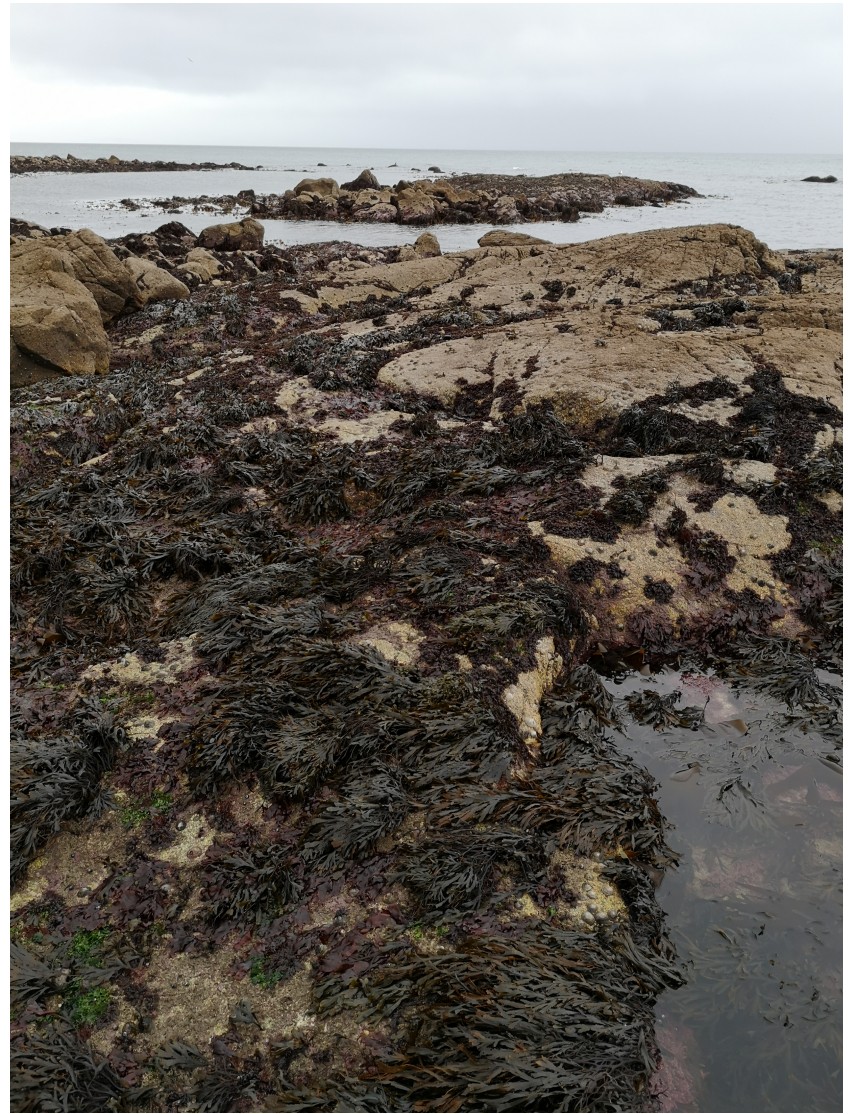

**Figure 1** **The lower shore at Furbo where sensors were placed.** Patches of dense algal cover occur adjacent to areas of open, barnacle-covered, rock.

*Fucus serratus* (Fig. 1). Half the loggers were placed in areas of 100% canopy cover with the rest placed in open areas without large fucoids. Loggers were interspersed, with two separate canopy patches (>25 m²) and two similar sized non-canopy areas used, with approximately 10 m between each patch. This arrangement nested two loggers separated by 2–3 m in each patch or open area. Average within patch correlations between logger measurements were similar to between patch correlations, so the hierarchical spatial structure was not explored further in subsequent analyses.

Average late summer wet weights of algae on this shore were 2,570 g m$^{-2}$ and 781 g m$^{-2}$ for *F. vesiculosus* and *F. serratus* respectively (*Johnson, 2020*). Canopy-forming fronds are typically between 0.5 and 1 m in length. Using a water content of 78.4% (*Stagnol et al.,*

*2016*), dry weight to area estimates from *Johnson et al. (1998)*, and only counting quadrats with a full canopy ($T_{AI}$ >1), the wet weights in Johnson (2000) suggest mean $T_{AI}$ of 4.55 (*F. vesiculosus*) and 3.07 (*F. serratus*) for the shore at Furbo.

Sensors recorded light (lux) and temperature (°C) at 30-minute intervals from March 2019 to February 2020. The response of the sensor tracks the standard sensitivity curve for human perception of light, with a Gaussian-type curve peaking at 550 nm and falling to no response for wavelengths outside of the 400 to 700 nm range. This peaked response contrasts with the idealized uniform response of PAR sensors in the same range. Engineering an ideal quantum yield between 400 and 700 nm partially explains the greater cost of PAR sensors. The HOBO sensors are factory calibrated to account for the plastic enclosure. The response to light at an angle is reported not to match the ideal cosine response: underestimating light landing on the sensor when the incident angle is between 30 and 80°. Potential sensor drift was investigated by examining the residual variance from a regression of average irradiance sensed in exposed loggers against daily global radiation from the Met Eireann station at Athenry (29 km from field location, http://www.met.ie).

The site was visited at approximately quarterly intervals and any organisms fouling the loggers were removed. In general, the loggers under canopies were clean, whereas exposed loggers were occasionally fouled by settling barnacles. Where the fouling occurred, it was more often around the rim of the logger than over the area where the sensor is located. The exposed conditions outside the canopy seemed to be more challenging for the loggers, with some failures during the year. The estimates of light without a canopy ($I_o$) were therefore based on the mean of measurements from open areas. Each logger under a canopy was treated as an individual timeseries of $I_c$ values. During the night, light measurements of zero cause the ratio $\ln(I_o/I_c)$ to be undefined, so $T_{AI}$ index values were treated as missing values in these cases. Logger data are available at https://doi.org/10.5281/zenodo.6797949.

Many algal canopies, such as those formed by *Fucus* spp., are relatively persistent. Growth is likely to be limited by processes like shading (*Middelboe, Sand-Jensen & Binzer, 2006*) with losses from senescence and seasonal storms not generally causing rapid loss of the entire canopy. Even species like *Sargassum muticum*, which have more pronounced senescence, have relatively gradual changes in biomass from month to month (*e.g.*, *Baer & Stengel, 2010*). The signal of canopy change was therefore estimated using a LOESS smoother. Using a LOESS smoother avoids the need to choose a date range to bin samples into. The span of the smoother sets the size of the moving window used for the smoother, with values between 0 (no smoothing) and 1 (all points). A span of 0.4 was chosen empirically as this proportion of the dataset gave curves that matched the assumed gradual build up and decline of biomass.

## RESULTS

Intertidal light intensity and temperature had seasonal cycles, along with considerable variability at shorter time scales (Fig. 2). Light peaked at an earlier date (day 164, June 13) than temperature (day 209, 28 July). Comparison of the smoothed values in successive winters suggests that 2020 had lower light intensity and cooler temperatures than 2019 at the start of the March.

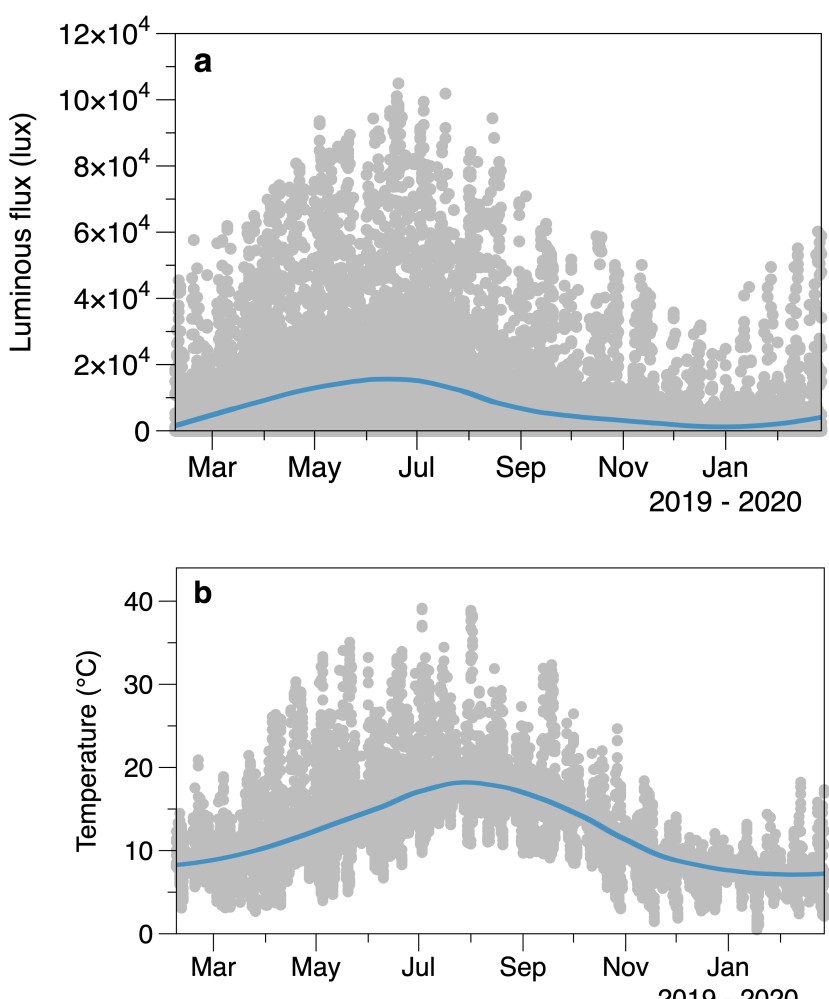

**Figure 2** **Average logged light (A) and temperature (B) from the sensors placed outside areas of algal canopy at Furbo.** Lines are fitted LOESS smoothers.

The higher frequency variation of light partly reflects the interaction of the times of tides with the day-night cycle (Fig. 3). Light both outside and under the canopy peaks in the middle of the day. Tidal fluctuations in water height also affect the light measured, with attenuation by the water column potentially causing flatter daytime peaks in light or even reducing light on the shore (*e.g.*, seen in exposed sensor light records between day 160 and day 164, Fig. 3)

Despite the differences in light measurement methods between the exposed HOBO sensors and meteorological data, there was a good relationship between the datasets (Fig. 4A). The lack of a significant slope in the residuals against date ($r^2$ 0.2%) suggests that the performance of the loggers did not drift over time (Fig. 4B).

Water depth did not affect the calculated $T_{AI}$ index from different canopy sensors. The range of values at water depths of 0 m (loggers exposed by tide and above water level)

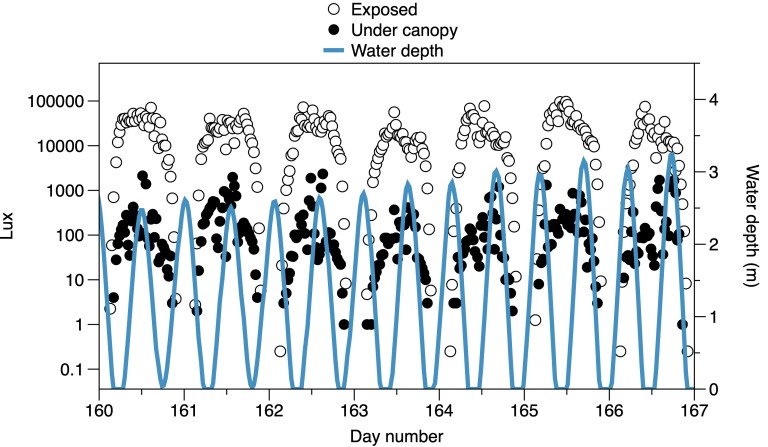

**Figure 3 Variability in light recorded from exposed sensors and sensors under the canopy, with tidal fluctuations in water depth above the sensors.** A 7-day period is shown, with midpoints indicating noon. The horizontal axis is centered on day 163.5: 12:00 on June 12 2019.

spanned the ranges of values seen when the loggers were submerged (Fig. 5A). When the time was expressed as a fraction of daylength, $T_{AI}$ index values showed a clear fall at times close to dawn and dusk (Fig. 5B). Subsequent $T_{AI}$ index measurements were therefore filtered to exclude records from the first tenth or last tenth of daylight.

The $T_{AI}$ index calculated from light measurements underneath and outside of canopies showed seasonal fluctuations (Fig. 6). Each under canopy sensor therefore described the expected seasonal variation in macroalgal biomass. Peak index values ($T_{AI}$ mean 6.0 SE 0.32) in the smoothed trend occurred around July 13th (day 194.5 SE 14.78). The lowest smoothed canopy index values were measured in February (mean day 51.0, February 20, SE 2.92) and averaged $T_{AI}$ 3.00 (SE 0.24). Consistent with the 2019 to 2020 light and temperature comparisons, all the smoothed trends in $T_{AI}$ index suggested higher and/or increasing biomass in March 2019, compared to lower and declining biomass at the end of February 2020.

## DISCUSSION

The seasonal changes in algal biomass observed using loggers match trends estimated for fucoids using harvests (*e.g.*, higher *F. vesiculosus* biomass in summer; *Attard et al., 2019*). These patterns are driven by higher growth rates in late spring/early summer (*e.g.*, *Stengel & Dring, 1997*) and periods of the highest net canopy production (*Bordeyne et al., 2020*). The timing and size of the seasonal peak in biomass are potentially driven by interactions between light availability, temperature, grazers, competition with epiphytes, reproductive allocation, frond erosion/breakage, and the dynamics of internal nutrient and carbon pools. *Graiff et al. (2020)* parameterize a model that includes most of these processes for a set of mesocosms: reproducing the observed seasonal growth variation.

A key issue for translating the $T_{AI}$ index into a biomass value are appropriate values for $k_a$, the attenuation by the algal canopy. Attenuation is likely to vary with species

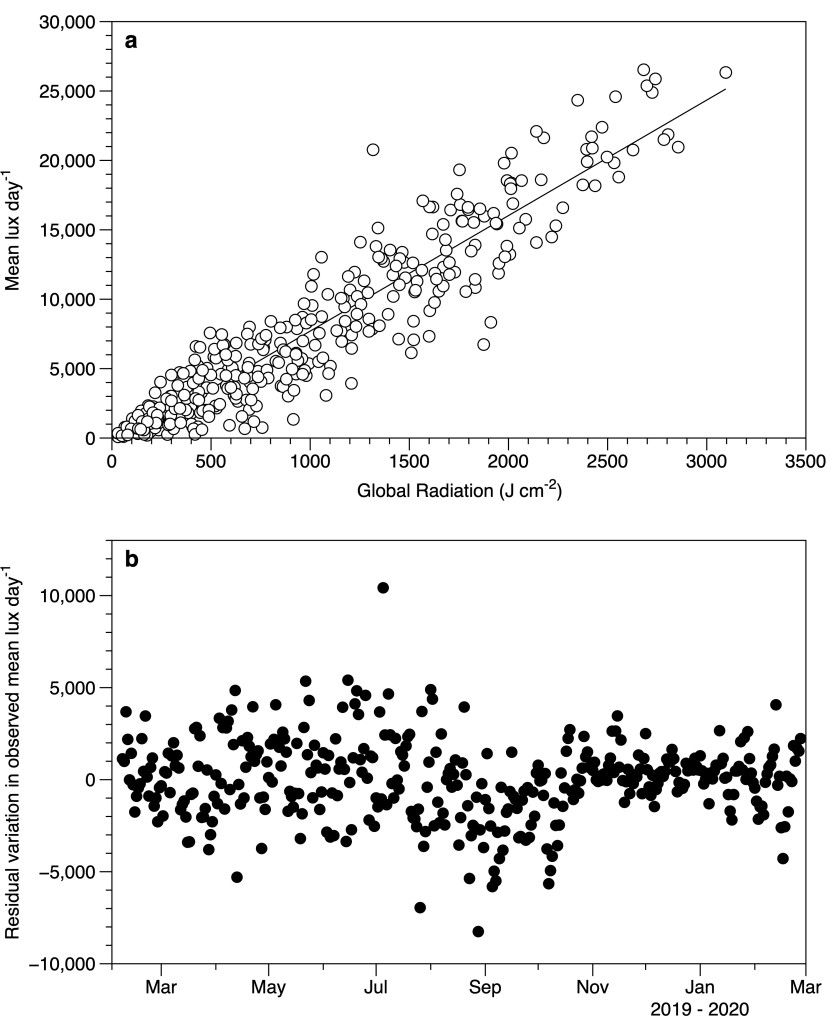

**Figure 4** **Light sensor intercomparison.** (A) Comparison between the average light intensity (lux) measurement for exposed sensors each day and the daily global radiation recorded at the Athenry meteorological station. The fitted line is a linear regression $r^2$ 88%. (B) Residuals from the regression in plot 4a as a function of date.

and wavelengths measured. For optically dark fronds like fucoids, there may be less wavelength dependence (*Dring, 1992*). However, there may be some seasonal variation in pigment content and morphology that could potentially affect $k_a$. Seasonal variation in the chlorophyll content of fucoids generally results in higher values during winter (*e.g.,* *Ruokolahti & Rönnberg, 1988*; *Stengel & Dring, 1998*). This might be expected to increase $k_a$, causing an overestimation of thallus area index in winter if a constant value of attenuation is used.

In general, there are few measurements of canopy attenuation. (*Johnson et al., 1998*) suggested a mid-range value of 0.7 ($T_{AI}{}^{-1}$). Applying a value of 0.7 as the canopy attenuation coefficient suggests seasonal variation in $T_{AI}$ values between 4.3 and 8.5: these values are broadly consistent with mean values for the canopy at Furbo based on wet weights (*Johnson,*
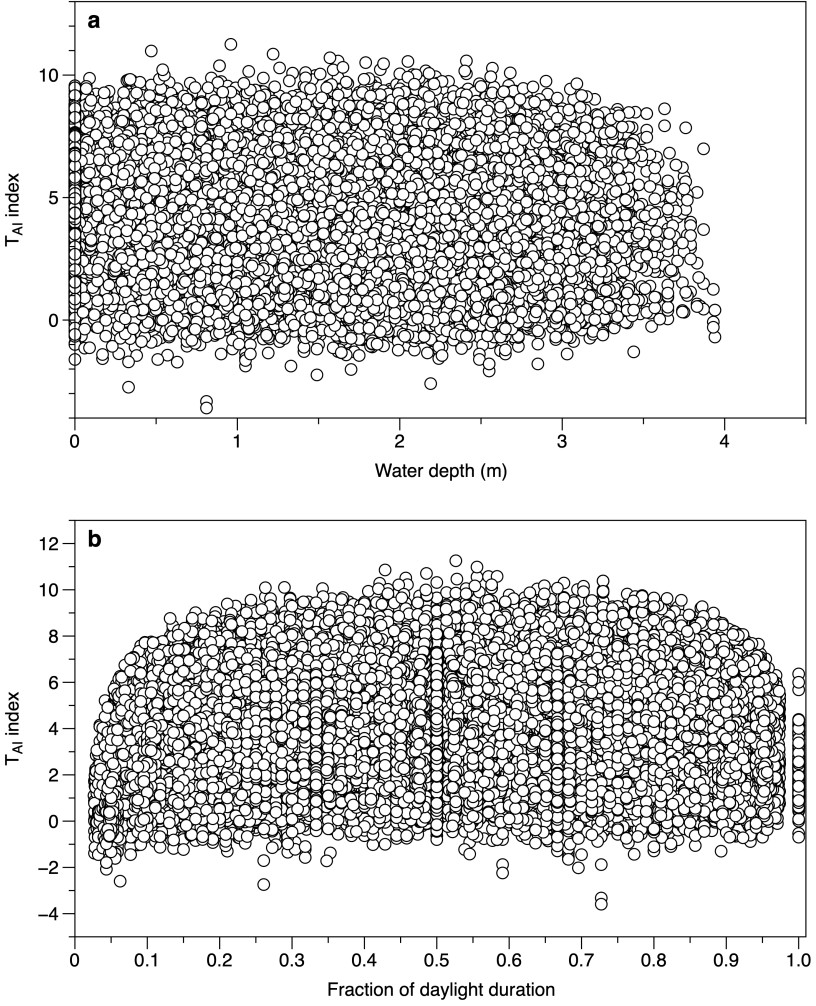

**Figure 5** **Environmental effects on the TAI index.** (A) All measurements of $T_{AI}$ index ($\ln(I_o/I_c)$) as a function of estimated water depth above the sensor. Values at a depth of 0 m are from sensors exposed to the air. (B) $T_{AI}$ index values at different times during daylight. The daylight duration was estimated for each day, with dawn scaled to 0 and dusk to 1.

*2020*). Clearly there is a significant gap in knowledge of macroalgal values of $k_a$, which requires a collaborative effort to define the scale of variability and important predictor variables for the extinction coefficient.

The loggers appeared to provide consistent measurements of light over the period of deployment, with measurements correlating with daily light measurements from a meteorological station. While water depth over the loggers did not have a clear impact on the $T_{AI}$ index, there was a crepuscular depression in index values. The deficiencies in cosine response of the sensors do not immediately explain this. Exposed sensors and those under the canopy might be expected to receive direct light at similar angles, so would both be subject to the underestimation effect. Possibly the scattering of light by the canopy altered the proportion of diffuse light (less affected by cosine issues) compared to exposed sensors.

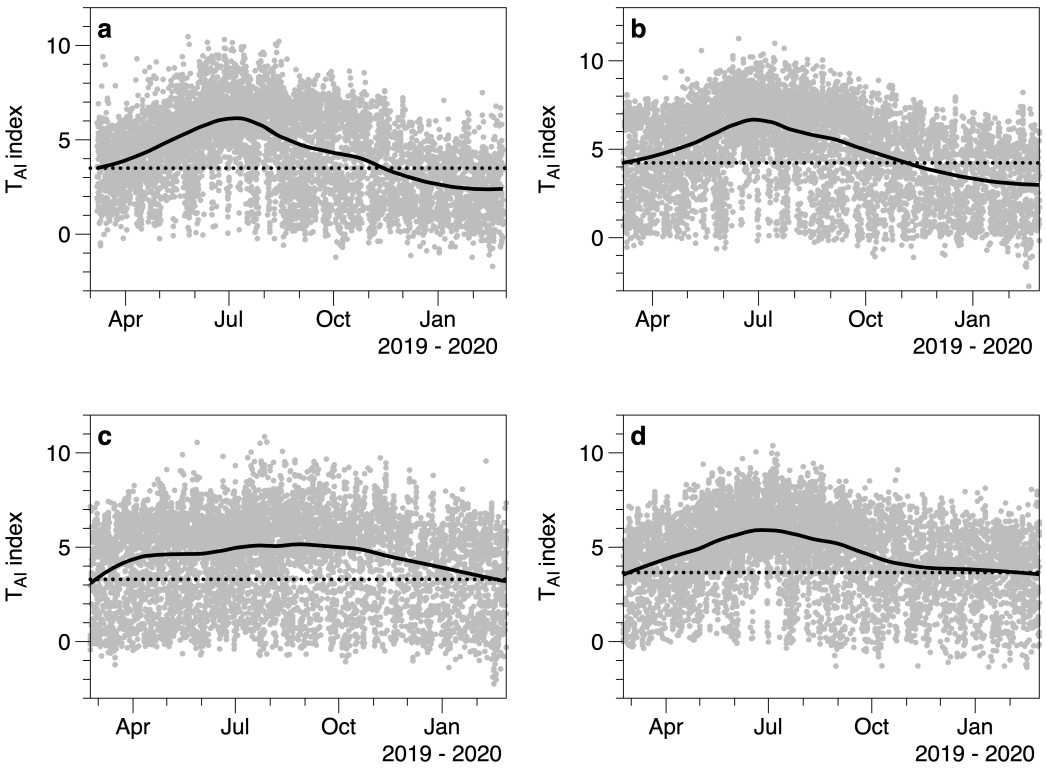

**Figure 6** **Values of the $T_{AI}$ index ($\ln(I_o/I_c)$) calculated for 4 different under-canopy sensors (A–D).** Solid lines are LOESS smoothers. Dotted lines reference the smoother value at the start of March 2019.

Alternatively, a decline in relative sensitivity of the sensors during the low light of dawn and dusk may reduce the ability to discriminate between shaded and exposed conditions, resulting in a decline in the $T_{AI}$ index.

If an appropriate canopy attenuation coefficient ($k_a$) can be defined, relative light attenuation seems to offer a practical, low cost means of monitoring algal biomass. In the absence of a suitable value for $k_a$, the method still supplies a proxy measurement of canopy biomass. Seasonal variations in biomass may indicate productivity, with the possibility to estimate interannual trends if sensors are maintained for longer periods. For example, the differences in canopy index values between March 2019 and February 2020 may indicate interannual variation and the pattern is consistent with the likely influence on photosynthesis of colder and darker weather in early 2020 compared to 2019 (Fig. 2).

More field measurements and comparisons will be needed to define the degree of inter species, intra species, and temporal variation in $k_a$. Different sensor types (e.g., PAR) will work as the method is based on a ratio rather than any particular units, however values of $k_a$ may vary with sensor type. The sensitivity of the method seems likely to decline at very high and low values of canopy biomass, for example if light is mostly not impeded by a canopy at low biomass. Sensors will also need a cleaning regime appropriate to local conditions if artefacts associated with fouling are to be avoided. The barnacle

fouling mentioned for Furbo was by settling cyprids in spring and early summer. If this affected the estimated biomass, the anticipated effect would be lower biomass: as $I_0$ was disproportionally reduced. No suppression of biomass or light recorded was apparent in spring and early summer (Figs. 4 and 6).

Relative light attenuation provides point estimates of canopy biomass (as $T_{AI}$ or $T_{AI}$ index). Further detailed modelling and observation may be needed to establish an appropriate spatial unit for the biomass estimate, assumed here to be 1 m$^2$. There may be underestimates of canopy biomass for light sensors on the edge of a patch, such that a proportion of the incident light reaches the sensor without canopy shading. Conversely, sensors in deeper water with larger algal species may be shaded by fronds several metres away.

## CONCLUSIONS

Relative light attenuation was used to derive seasonal cycles of biomass consistent with the expected seasonal fluctuations of fucoid algae. Despite the noisy signal, the approach offers a low cost means of monitoring algal biomass without the disadvantages of destructive harvests.

The light attenuation method can complement existing programmes for a low cost. Different types of waterproof light sensors are suitable, with additional possibilities for scheduling monitoring or sensor maintenance if data are sent continuously by modem or satellite. The major benefits of the method are that a direct proxy for biomass is non-destructively measured at time intervals sufficient to follow seasonal processes. Longer time series can be established at relatively low cost, given the constraints of needing to replace or clean sensors periodically. Light loggers are widely available and established technology, it is possible that many existing datasets could be reanalysed to provide biomass proxies in cases where loggers have been placed under canopies.

As loggers provide point estimates of canopy biomass, the relative light attenuation method would be most informative when used alongside methods which complement this. For example, other techniques, like remote sensing, are more suitable for estimating the spatial extent of canopy cover. Destructive surveys provide unambiguous canopy biomass estimates but using relative light attenuation methods can reduce the frequency of harvests and provide information on canopy biomass between harvests. While remote sensing can provide estimates of biomass, different images may need seasonal corrections and there may be a need for image-by-image adjustments. Relative light attenuation can provide an independent means to constrain or cross check remote sensing methods.

### Funding
The author received no funding for this work.

### Competing Interests
The author declares that he has no competing interests.

## Author Contributions

- Mark Johnson conceived and designed the experiments, performed the experiments, analyzed the data, prepared figures and/or tables, authored or reviewed drafts of the article, and approved the final draft.

## Data Availability

The data is available at Zenodo: mar-env. (2022). mar-env/Canopy-attenuation: Dataloggers_Furbo (v1.0). Zenodo. https://doi.org/10.5281/zenodo.6797949.

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
