# Peer review of "Low cost macroalgal canopy biomass monitoring using light attenuation"

_PeerJ, doi:10.7717/peerj.14368_

## Round 0.1 · original submission · Major Revisions

Both reviewers think this approach is potentially useful, and could benefit from some additional analyses, such as validation with additional data.

Reviewer 1 ·

Basic reporting

This manuscript describes a novel approach to estimate the macroalgal biomass at intertidal sites based on the light attenuation caused by various amounts of algal thalli. The metho appears sound, and the author describes in detail the advantages of the methods and identifies its potential problems. High resolution data, collected during a whole year, are presented to illustrate the potential of the method.
As is, I cannot recommend that the paper is accepted, suggesting major revision and ideally the collection of more data.

However, the work seems incomplete, especially because no data is given about the key parameter, k. This parameter is critical to allow the estimate of biomass from light measurements. Maybe the author could attempt to estimate k, maybe on separate, dedicated, laboratory tests?

Experimental design

Another important aspect to validate the method would be to have direct comparisons between estimates and actual independent measurements of macroalgal biomass or cover. As mentioned in L213 and onwards, the only thing that can be said about the validity of the method is that the seasonal variation of the index TAI is compatible with very general seasonal patterns known to occurred.

Validity of the findings

Regarding the figures:
1. Results are focused on seasonal variability but would be interesting to see the data with more temporal detail, for shorter time scale, for example 1 day, 1 fortnight cycle. It would reenforce the value of having high frequency measurements.
2. I would love to see the measurements of light under clear water and under the canopy side by side, before being integrated in the index TAI.
3. Also, it would help the reader if the months were shown in the x-axis

Additional comments

Minor comments:

L108, 145 noise: can these (or other) sensors integrate over time?

L149: Select specific wavelength bands, related to chl absorption (blue, red)?

L165: How far apart were the open and shaded locations?

L173: What is the spectral sensitivity of these sensors?

L174: ‘quarterly’? Any evidence of gradual decrease in signal due to biofilms, epiphytes?

L181. When and why did this happen?

L189-193. Please explain with more detail this choice.

L231 ‘can’?

L231. K may also vary along time (season), as thalli become thicker and darker, or as some species are replaced by others.

Reviewer 2 ·

Basic reporting

The English language is clear and professional. Literature references are mostly sufficient, although I note one instances in Additional Comments where I felt additional references were appropriate.

The author present the raw data files, but I had difficulty interpreting the data without the satisfactory explanations. For example, it was not clear which sensors (in and out of canopy) were supposed to be paired (i.e., which were nearby and at the same depth). If I assumed that sensor NC1a corresponds to sensor NC2a, then the majority of those observations appear to not have coincident measurements. Can the author please clarify this? And if there are not coincident measurements, I was unclear why Figure 2 shows consistent time windows for all paired measurements. I think that manuscript would be improved with better clarity in the presentation of raw data.

The article structure is professional and appropriate, although the results section appears significantly underdeveloped, and the article could be significantly improved by presenting more information and analyses in the results. I will try to suggest some additions in the experimental design section.

The article is self-contained.

Experimental design

The author describes a novel approach for monitoring relative changes in macroalgae based on comparing light loggers placed inside and outside of known algal patches, and the author presents a dataset collected in the mid to low intertidal zone to demonstrate the application of this approach .

I believe the methods would be improved by clarifying more information about the instrument and the biology in the study.

For example, related to the instrument, I think it would be useful to note:
What are the spectral characteristics of the Hobo logger?
How do these spectral characteristics relate to the spectral dependencies of the macroalgae, and how do they relate to the water types commonly observed in this region?
What are the noise levels and sensitivities of the light logger, particularly related to the high number of zeros in the dataset?
What is the acceptance angle / cosine response of the light logger, since the site is high latitude, in regards to diffuse versus direct contributions?

Related to the biology and region:
What were the observation depths?
What are characteristic heights of the fucoid algae?
How does the author expect self-shading of the fucoid algae to affect measurements?

Validity of the findings

The approach proposed by the author appears feasible based on the equations presented, but the article lacks validation data needed to evaluate the data collected using the author's approach. If validation using an independent dataset is not possible, then I believe that additional analyses are be needed to determine whether the author's data supports the proposed approach.

For example, the dataset should be compared to tidal swing and diurnal cycles.
Does the ratio ln(Io/Ic) show correlation to the tidal record or to the diurnal cycle? The author notes on line 152-154 that tidal depths and variations in seawater attenuation "are likely to be minor", but does not conduct analyses to support this. When I attempted to use the dataset provided, a preliminary viewing suggested that the daily solar cycles were not cancelled out in the ln(Io/Ic) ratio. If this isn't clarified, then it's not clear that the subplots in figure 2 aren't depicting changes related to solar geometry or other effects.

The author could also have analyzed high resolution imagery to try to provide support for the timing of the summertime peak determined by the light loggers.

Additional comments

I think the author has done a very nice job of describing a method and presenting a dataset, but that additional analyses are required to evaluate the method. I've listed some additional line comments below:

L50: Does cover at any of the author's sites "exceed 100%", since the author mentions this is a challenge for other methods?

L94: Bell et al 2020 and others have presented advances in satellite automation, which should be noted here.

L109: "evaluates whether data logging can provide a robust description of summertime peaks" There is no validation data to evaluate the methods shown herein, so I'd recommend that this language is modified.

L120: Is it common to use lambda to refer to an attenuation coefficient? I'm more familiar with attenuation being described by a variable K, with lambda indicating wavelength, and a subscript indicating the material, i.e., Kw is the attenuation of pure seawater.

L139: What are the dependencies of K, e.g., species, morphotype, and how is K affected by self-shading?

L163: An image of the field site would be very helpful. If the author does not have an image already, a scene from a high resolution satellite would be very helpful to orient the reader. A map could suffice.

L169-171: Are the biomass calculations ever performed for this article's dataset?

L181: "Light measurements of zero" What is the sensitivity/noise of the logger that led to these null measurements? Also, do the loggers ever get exposed to air during tidal swings? How do the wide changes in conditions (e.g., temperature) alter the logger's calibration, and is any calibration tracking performed?

L183: Were calibrations performed beginning and after deployment? Did the sensor baseline shift? For example, 2 of 4 timeseries in Figure 2 end below the initial value, and none end higher. How does macroalgal cover compare between the start and end of observations? The author noted that fouling was more common for light loggers outside of the canopy. If those outside loggers were to decrease their responsivity over the course of the field survey, wouldn't this also cause downward trends in T_AI index?

L190: Given the substantial noise, what is the uncertainty in the timing of the peak in the LOESS estimate?

L191: "A span of 4" Does this refer to 4 days? Are units helpful for interpreting this?

L251 vs L 259: There seems to be some ambiguity in the manuscript between whether the dataset has been used to predict cover or biomass. Although the author describes that cover can be a proxy for biomass, I think clarity of language on this topic could be improved.

---

## Round 0.2 · accepted · Accept

Thanks for the changes to the paper -- the reviewer understands additional data collection isn't possible but is otherwise satisfied with the changes made in the last revision. Congratulations!

Reviewer 1 ·

Basic reporting

No comment.

Experimental design

No comment.

Validity of the findings

No comment.

Additional comments

I understand that the collection of more data is impossible at this stage, and I accept the arguments of the author and the alternative option of analyzing the results in more detail. Together with the new figures, the additional information provided, and various edits, I think the manuscript was improved substantially can be accepted for publication.